# Are Atmospheric Updrafts a Key to Unlocking Climate Forcing and Sensitivity?

Leo J. Donner[1], Travis A. O'Brien[2], Daniel Rieger[3], Bernhard Vogel[3], and William F. Cooke[4]

[1]NOAA/GFDL, Princeton University Forrestal Campus, 201 Forrestal Road, Princeton, New Jersey 08540 USA
[2]Lawrence Berkeley National Laboratory, Berkeley, California USA
[3]Karlsruhe Institute of Technology, Karlsruhe, Germany
[4]UCAR/GFDL, Princeton, New Jersey USA

*Correspondence to:* L.J. Donner (leo.j.donner@noaa.gov)

**Abstract.** Both climate forcing and climate sensitivity persist as stubborn uncertainties limiting the extent to which climate models can provide actionable scientific scenarios for climate change. A key, explicit control on cloud-aerosol interactions, the largest uncertainty in climate forcing, is the vertical velocity of cloud-scale updrafts. Model-based studies of climate sensitivity indicate that convective entrainment, which is closely related to updraft speeds, is an important control on climate sensitivity.
Updraft vertical velocities also drive many physical processes essential to numerical weather prediction.

Vertical velocities and their role in atmospheric physical processes have been given very limited attention in models for climate and numerical weather prediction. The relevant physical scales range down to tens of meters and are thus frequently sub-grid and require parameterization. Many state-of-science convection parameterizations provide mass fluxes without specifying vertical velocities, and parameterizations which do provide vertical velocities have been subject to limited evaluation
against what have until recently been scant observations. Atmospheric observations imply that the distribution of vertical velocities depends on the areas over which the vertical velocities are averaged. Distributions of vertical velocities in climate models may capture this behavior, but it has not been accounted for when parameterizing cloud and precipitation processes in current models.

New observations of convective vertical velocities offer a potentially promising path toward developing process-level cloud
models and parameterizations for climate and numerical weather prediction. Taking account of scale-dependence of resolved vertical velocities offers a path to matching cloud-scale physical processes and their driving dynamics more realistically, with a prospect of reduced uncertainty in both climate forcing and sensitivity.

## 1 Introduction

Uncertainties in both anthropogenic climate forcing and climate sensitivity continue to limit our understanding of climate
change and the precision with which scenarios for future climate change can be constructed. As had Kiehl (2007) for an earlier generation of climate models, Forster et al. (2013) found that CMIP5 models able to successfully simulate observed global warming over the pre-industrial to present-day period did so by balancing a range of anthropogenic climate forcings and climate sensitivities. Both forcing and sensitivity have proved resistant to reducing their uncertainties. Understanding the

relative roles of forcing and sensitivity, along with variability, is essential to actionable estimates of future climate change. Given the long lifetime of greenhouse gases relative to atmospheric aerosols, aerosol "masking" of warming by greenhouse gases over the pre-industrial to present-day period will become less important. The future corresponding to a "low net-forcing, high-sensitivity" twentieth century differs from that for "high net-forcing, low sensitivity." The importance of this question

is cast into stark relief by assessments of the extent to which the Intended Nationally Determined Contributions, emissions reductions pledged at the recent Paris COP21 conference, will meet the COP21 goal of holding increases in globally averaged temperature below $2^{\mathrm{o}}$ C by 2100. The relationship between temperature increases, other elements of human-induced climate change, and emissions remains uncertain within a range implied by ongoing uncertainty in sensitivity. Reducing this uncertainty would be of great value as emissions goals are revised going forward.

This review presents the perspective that the vertical velocities, updrafts on all scales, are among the keys to understanding and simulating climate forcing and are plausibly also important for climate sensitivity. Vertical velocities have received limited attention in climate (and even cloud) models and have only recently become a focus of observational studies. Sub-grid clouds are parameterized in climate models, traditionally based on mass fluxes (product of vertical velocity, density, and area) without vertical velocity specifically. The dependence and realism of explicitly resolved vertical velocities, even in higher-resolution

models, has not been extensively examined.

## 2   Vertical Velocity and Climate Forcing

Aerosol-cloud interactions are the largest source of uncertainty in climate forcing, with estimates ranging from close to zero to -1.3 W m$^{-2}$, in contrast to forcing by carbon dioxide of $1.7 \pm 0.4$ W m$^{-2}$(Stocker et al., 2013). Rosenfeld et al. (2013) and Rosenfeld et al. (2014) discussed elements of aerosol-cloud interactions leading to this uncertainty. Here, we emphasize

that the updraft speeds at which cloud liquid and ice are activated are among the primary controls on cloud drop sizes and number concentrations, which are in turn related to cloud optical properties, precipitation, and macrophysical properties. Fig.1 shows the sensitivities of drop effective size (Feingold, 2003) and number concentration (McFiggans and Co-Authors, 2006) to vertical velocity, aerosol number concentration and size distribution, and aerosol composition. Drop sizes and number concentrations are more sensitive to vertical velocity than aerosol composition, and, as aerosol concentrations increase from

clean to polluted, vertical velocity becomes increasingly important relative to aerosol number and size. For homogeneous freezing, ice crystal number concentrations are often controlled more by vertical velocity than aerosol number concentration (Fig. 2, (Kay and Wood, 2008)).

In general, then, physically based simulation of aerosol-cloud interactions requires knowledge of the updraft speeds at which these interactions occur, along with requisite information on aerosol composition and size distributions. It is important to note

that these updrafts occur on a wide range of scales, down to large eddies with horizontal and vertical scales of tens of meters. As a matter of simulation, this implies parameterization of vertical velocities in models whose scales are coarser than models

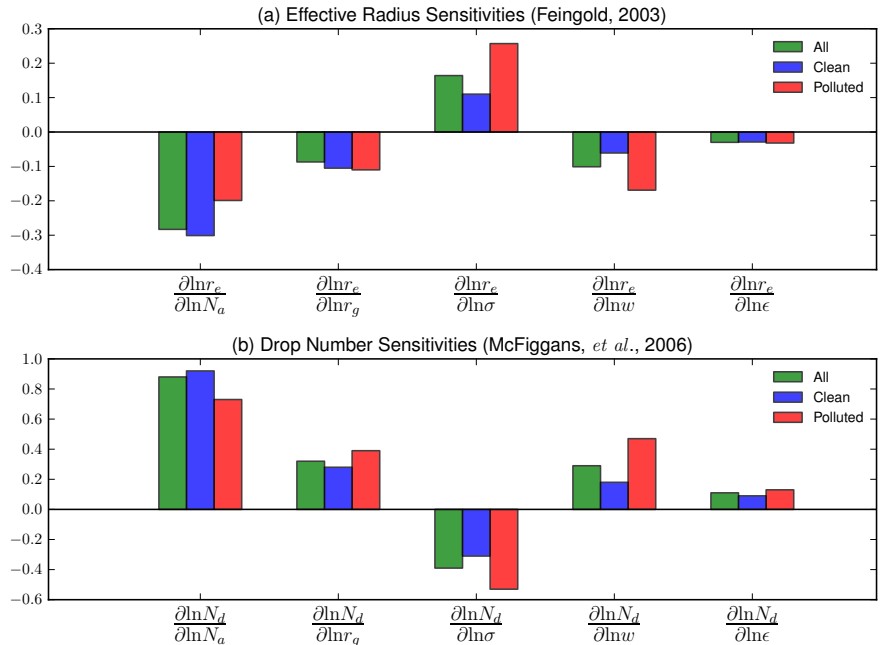

**Figure 1.** Sensitivities of (a) drop effective radius ($r_e$) (Feingold, 2003) and (b) drop number concentration ($N_d$) (McFiggans and Co-Authors, 2006) to aerosol number concentration ($N_a$), aerosol median size ($r_g$), breadth parameter ($\sigma$) for the uni-modal log-normal distribution assumed for aerosol sizes, vertical velocity ($w$), and aerosol mass fraction ($\epsilon$) of ammonium sulfate. The drop effective radius $r_e$ is the ratio of the number-weighted radius cubed to the number-weighted radius squared.

that resolve large eddies, and, even in large-eddy simulations, attention to the realism of distributions of vertical velocities will be important.

## 3 Vertical Velocity and Climate Sensitivity

The possible relationship of vertical velocity to climate sensitivity, broadly defined as the response of a climate measure such as global, annual-mean surface temperature to a change in climate forcing, is less obvious. The strongest suggestions of a link emerge from several studies showing that convective entrainment, an important control on vertical velocity, is related to the climate sensitivity in general circulation models (Stainforth et al. (2005), Rougier et al. (2009), Sanderson et al. (2010), Zhao (2014)). This result is perhaps surprising, given that low- and mid-level-cloud feedback is the major uncertainty in climate sensitivity (Zelinka et al., 2012). One possible explanation is provided by Sherwood et al. (2014), who trace climate sensitivity to convective mixing, in turn related to dehydration of low-cloud layers. In their perturbed parameter experiments with a climate model, Klocke et al. (2011) found that the entrainment rate for shallow convection explained most of the variation in its sensitivity, with no sensitivity to entrainment rate for deep convection. As a mechanism for forming low clouds and an agent

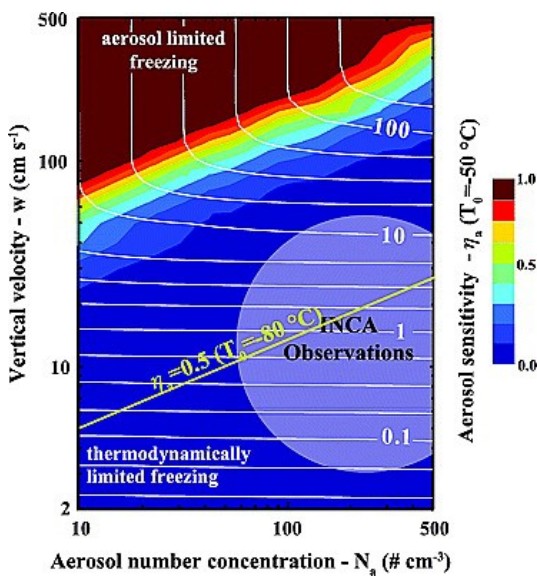

**Figure 2.** Aerosol sensitivity (derivative of logarithm of ice crystal number concentration with respect to logarithm of aerosol concentration) and ice number concentration (contoured) as functions of vertical velocity and aerosol number concentration for homogeneous freezing, from Kay and Wood (2008). Shaded aerosol sensitivities are for a parcel lifted at $-50^{\circ}$C and 250 hPa, with the line indicating an aerosol sensitivity of 0.5 at $-80^{\circ}$C and 100 hPa. Observations (10-90% percentile) from the INCA (INterhemispheric differences in Cirrus properties from Anthropogenic emissions) field campaign fall within the light circle.

for dehydrating the lower atmosphere, shallow convection quite plausibly is important for climate sensitivity. Heat released in deep convection drives the Hadley and Walker circulations, and, as noted by Su et al. (2014), changes in cloud radiative effect in the tropics are closely related to patterns of strengthening and weakening of the Hadley circulation. Taken together, these results point to the important role played by deep cumulus convection in determining the dynamic and thermodynamic environments in which stratocumulus and shallow cumulus form. Though low and mid-level clouds are the proximate agent determining climate sensitivity, deep convection is among their remote controls and thereby also possibly important to climate sensitivity. The roles of these mechanisms likely exhibit model dependence, as suggested, for example, by Klocke et al. (2011)'s insensitivity to

deep convective entrainment. Entrainment and detrainment in convection are closely related to their vertical profiles of mixing, heating, moistening, and drying. Through its influence on buoyancy, entrainment bears a close relationship to vertical velocity. Though less directly tied to climate sensitivity than climate forcing, vertical velocities through these mechanisms are quite plausibly correlated to climate sensitivity.

The mechanisms discussed above explore fundamental characteristics of convection (convective mixing with associated dehydration of low-cloud layers, shape and vertical extent of convective heating and moistening, convective microphysics, interactions between convective and stratiform precipitation) and their possible relationships to climate sensitivity. Vertical velocity does not directly relate to climate sensitivity, but, rather, correlates with these characteristics and is an indicator of how they are functioning in the climate system. As observed vertical velocities become available at convective scale, they
thereby provide an important, previously unrealized, constraint on these processes.

## 4    Modeling Implications and Prospective Breakthroughs

The implications of strong dependencies of climate forcing and sensitivity on vertical velocities present a challenge to current climate model development but also hold the promise of possible breakthroughs. Since the physically relevant scales for vertical velocity are not only resolved explicitly in climate models but also sub-grid, both the vertical velocities from the model
dynamical cores and those in parameterizations for sub-grid processes are important. Neither has been given much attention in model development to date. Indeed, many parameterizations of sub-grid processes, e.g., mass-flux parameterizations for cumulus convection, have not even provided vertical velocities. Therein lies the promise for possible breakthroughs, as attention can be turned to these issues. There are numerous challenges, related especially to scale awareness for physical processes and realism of parameterized and resolved vertical velocities.

Empirically and theoretically, there is strong reason to suspect that vertical velocity should depend on resolution (Rauscher et al., 2016; O'Brien et al., 2016). Consider the discretized continuity equation: $\Delta_x u_D / \Delta x + \Delta_y v_D / \Delta y + \Delta_p \omega / \Delta p = 0$, where $u_D$ and $v_D$ represent the horizontally divergent wind components, $\omega$ represents vertical velocity in pressure coordinates, $\Delta_*$ represents a finite difference operator, and $\Delta x \sim \Delta y$ represents the horizontal differencing distance. Scale analysis of this equation implies that vertical velocity scales as $|\omega| \propto |\Delta_x u_D| / \Delta x$. Inspection of this relationship reveals that $|\Delta_x u_D|$ is
equivalent to the first order structure function of wind, which has been demonstrated to exhibit power-law behavior in nature: $|\Delta_x u_D| \propto x^H$ (e.g., Cho and Lindborg, 2001). Hence the combination of mass continuity and the scaling properties of the wind field imply that vertical velocities should change with averaging distance or, in a model, resolution: $|\omega| \propto \Delta x^{H-1}$. For typical structure function exponents of $H \sim 1/3$, this implies that vertical velocity increases with resolution like $\Delta x^{-2/3}$. Fig. 3 shows that the structure function from a high-resolution forecast model indeed closely follows this $\Delta x^{-2/3}$ relationship over
the range of resolutions represented in the CMIP3 and CMIP5 model archives. At considerably smaller, non-hydrostatic scales, the nature of the scaling may change. Pauluis and Garner (2006) report that updraft speeds in a non-hydrostatic model scale with the ratio of grid size to updraft vertical extent for resolutions finer than 16 km.

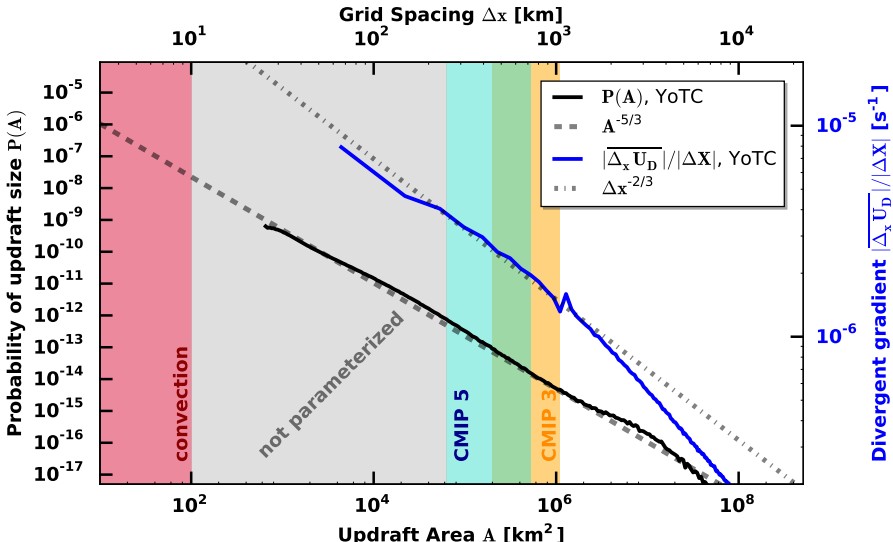

**Figure 3.** Probability distribution of updraft area $P(A)$ (black curve; left axis) and the structure function of the zonal, divergent wind component $\overline{|\Delta_x U_D|}$ divided by the structure function distance $\Delta x$ (blue curve; right axis). Both $P(A)$ and the structure function are derived from 1.5 years of 500 hPa output from the ECMWF Year of Tropical Convection (YoTC) T799 operational forecast output (European Centre for Medium-Range Weather Forecasts, 2012). The shaded golden and blue regions correspond to the inter-quartile range of CMIP3 and CMIP5 models respectively, and the shaded rose region corresponds to the scales at which convection occurs. $A^{-5/3}$ and $\Delta x^{-2/3}$ power laws are provided for reference (gray dashed and gray dash-dotted lines respectively).

Relatedly, the probability distribution of updraft sizes in the same forecast output also closely follows a power law of updraft area over all observable scales in the model (Fig. 3). However, the current paradigm for the subgrid representation of updrafts assumes a separation of scales: that convection occurs at scales $\mathcal{O}(10km)$ or less. This implies that there is a continuum of updraft sizes below the resolution of contemporary climate models that are not represented by current parameterizations.
5   Increasing model resolution will increase the representation of updrafts in the unparameterized continuum, represented by the gray swath in Fig. 3. So unless subgrid parameterizations compensate accordingly, this will naturally result in an increase in the vertical mass fluxes as resolution increases.

As discussed previously, vertical velocity is a critical control on droplet and ice activation. This suggests that climate forcing could exhibit a dependence on model resolution, even if vertical velocities were simulated realistically for a given resolution.
10   Stevens (2015) relates climate forcing $F_{aci}$ by aerosol-cloud interactions to anthropogenic changes in cloud drop number $N_d$:

$$F_{aci} = -CE\left(\frac{\delta N_d}{N_d}\right), \tag{1}$$

where $CE$ is the effective cloud fraction. All quantities are global, annual means. A very rough estimate of the effect of the resolution dependence of vertical velocity can be made assuming $CE$ is fixed and that the partial derivatives shown in Fig. 1 do not co-vary significantly, so that $\delta N_d$ is also fixed. For a resolution refinement from $2^{O}$ to $0.25^{O}$, the vertical velocity
15   scalings in Fig. 3 and the values for the variation of drop number with vertical velocity in Fig. 1 imply reduction in $F_{aci}$ of

about 18% to 36%, due to increases in $N_d$ of about 22% to 56%. Larger vertical velocities increase the pre-industrial droplet number $N_d$ and reduce the sensitivity of clouds to the number perturbation $\delta N_d$. Ma et al. (2015) found that aerosol indirect forcing decreased by about 30% in the Northern Hemisphere mid-latitudes, where most anthropogenic emissions occur, and 15% globally in CAM5 as horizontal grid spacing was decreased from $2^\text{o}$ to $0.25^\text{o}$. Resolved vertical velocities are among the factors to which aerosol indirect forcing is related in their study. Aerosol forcing in a comprehensive model like that used by Ma et al. (2015) depends not only on droplet and ice nucleation and their relationships to vertical velocity. Ma et al. (2015) attributed their resolution sensitivities for aerosol indirect forcing to the resolution dependencies of their parameterizations for droplet nucleation and precipitation. Despite this, the resolution sensitivity of forcing found by Ma et al. (2015) agrees well with estimates based on (1).

Zhang et al. (2016) examined an important component of aerosol indirect forcing, the sensitivity of liquid water path to the concentration of cloud condensation nuclei. They found this sensitivity to vary with dynamical regime within models and across models for individual regimes. Zhang et al. (2016)'s study is consistent with an important control on aerosol-cloud interactions by vertical velocities, which change with dynamical regime and, in all likelihood, within regimes across models.

To pursue the question of resolution dependence of a specific microphysical process related to aerosol indirect effects, we consider the dependence of ice activation on model resolution. We modeled ice activation by heterogeneous freezing in a massive Saharan dust plume which advected to central Europe on 3 April 2014 using the ICON (ICOsehedral Non-hydrostatic) numerical weather prediction and climate model (Zängl et al., 2015), developed at the Deutscher Wetterdienst and Max Planck Institute for Meteorology and extended as ICON-ART to include aerosols and their interactions with clouds (Rieger et al., 2015). Fig. 4a shows the vertical velocities at which heterogeneous freezing occurs depend strongly on the ICON-ART horizontal resolution. Figs. 4b-d show that this dependence leads to large changes in the number of ice particles produced by heterogeneous freezing, with overall increases in ice crystals formed at finer resolutions. These results show the distributions of cloud microphysical, and thereby radiative and precipitation, properties depend on model resolution. The power-law dependence of the observed structure functions, shown in Fig. 3, suggests that such dependencies could possibly be taken into account by scaling the vertical velocities used to calculate activation with model resolution. Cusack et al. (1999) exploited this scaling to estimate unresolved variance of saturation in developing a cloud parameterization.

Updraft scales extend down to large eddies (tens of meters), so modeling cloud processes depending on vertical velocities requires their sub-grid scales to be parameterized. In climate models, resolutions are still coarser than convective scales, leaving the dependence of climate forcing and sensitivity on convection also to be parameterized. Realistic vertical velocities for parameterized convection would satisfy an important constraint related to entrainment, detrainment, and convective mixing. Many cumulus treatments in climate models parameterize only mass fluxes and do not provide its factors (vertical velocity, area, and density) independently, but some cumulus parameterizations do, e.g., the parameterizations for shallow (Bretherton et al., 2004) and deep (Donner, 1993) convection in the Geophysical Fluid Dynamics Laboratory Climate Model-3 (Donner et al., 2011). Even for these parameterizations, limited attention has been directed to the realism of their vertical velocities.

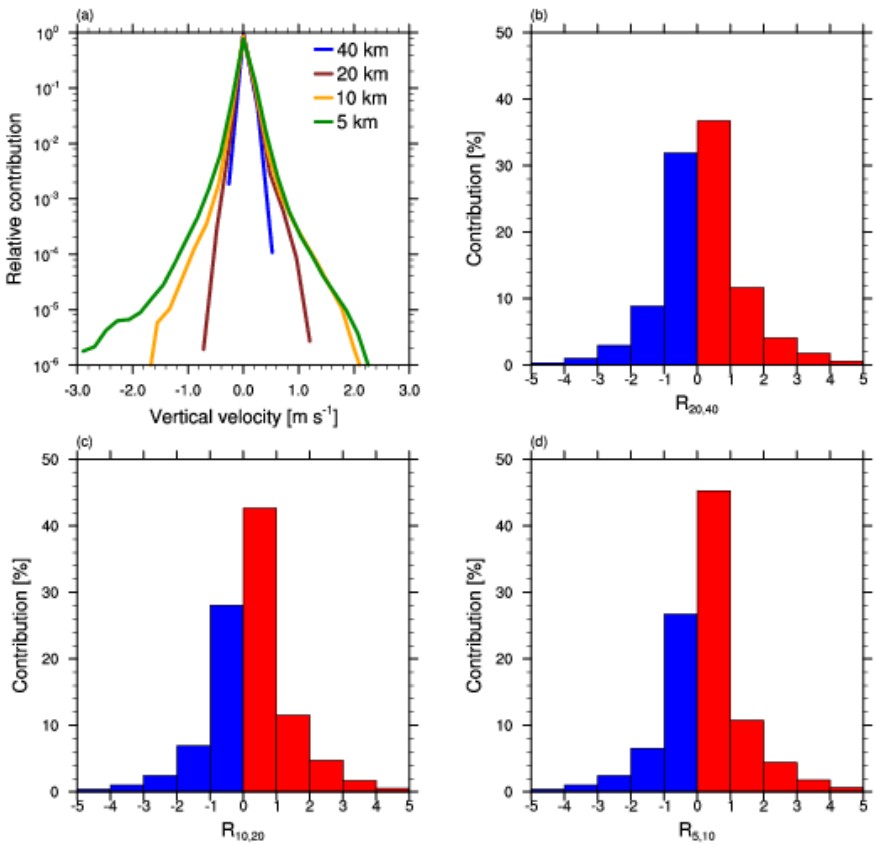

**Figure 4.** (a) Relative frequency of vertical velocity in the heterogeneous freezing regime (235 K < T < 273.15 K) for ICON-ART horizontal resolutions of $\Delta x = 40, 20, 10,$ and 5 km. (b)-(d) Frequency distribution of the ratio of the formation rates of ice crystal number produced by heterogeneous freezing. The ratio is defined as $R_{\Delta x_2, \Delta x_1} = \log(\bar{n}_{het,\Delta x_2}/n_{het,\Delta x_1})$ with the number of formed ice particles at coarser resolution, $n_{het,\Delta x_1}$, and the mean number of formed ice particles in the corresponding spatial volume (model grid cells) at finer resolution, $\bar{n}_{het,\Delta x_2}$. Values of $R_{\Delta x_2,\Delta x_1} > 0\,(< 0)$ indicate an increase (a decrease) in ice crystal nucleation with an increase in resolution. The results are for the period 12 UTC, 3 April 2014, to 0 UTC, 4 April 2014, for a circular domain with a radius of $8^O$ centered at $6^O$W, $46^O$N. ICON-ART was initialized with a 15 March 2014 European Centre for Medium-Range Weather Forecasts Integrated Forecast System analysis to spin-up background dust, with daily forecasts from 29 March until 4 April. Domains ranged from global for 40-km resolution to central Europe for 10 and 5-km resolutions.

Donner (1993)'s parameterization was calibrated using observations of convective vertical velocities from the Global Atmospheric Research Program Atlantic Tropical Experiment (GATE) in the 1970s. Only recently have observations from other field campaigns permitted independent evaluation of the manner in which vertical velocities are calculated in that parameterization. Fig. 5 shows the parameterization captures the basic shape of the updraft profile observed in the Tropical Warm

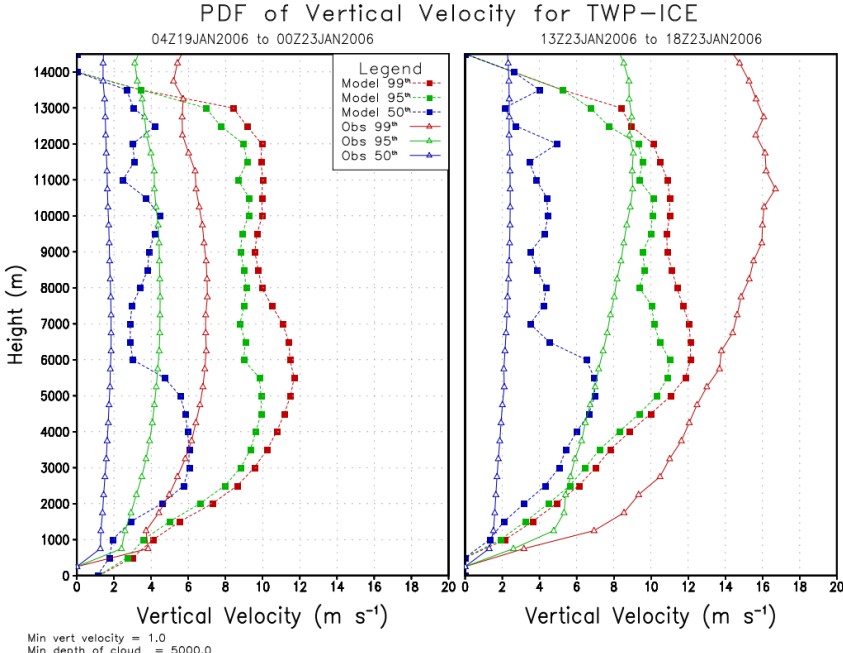

**Figure 5.** Probability distribution functions of vertical velocities in deep convective updrafts using the cumulus parameterization in Donner et al. (2011) and observed using dual-Doppler radar (Varble et al., 2014). The panels show two periods (04 UTC 19 January to 00 UTC 23 January 2006 and 13 UTC to 18 UTC January 2006) for which radar observations are available, along with model-generated updrafts for the same periods. Both panels show radar-observed and model-generated updrafts whose vertical extent is at least 5 km with a minimum speed of 1 m s$^{-1}$ or larger. The percentiles (colors) are the fraction of the updrafts with vertical velocities less than the plotted velocities, as functions of height.

Pool-International Cloud Experiment (TWP-ICE). The updraft speeds for the strongest updrafts are generally within a factor of two of observations with the parameterized updrafts stronger than observed in one case and weaker in the other (Varble et al., 2014). Consistent with radar observations, the modeled median vertical velocities are similar over both time periods analyzed, but the observed strongest 1% of the vertical velocities differ by about a factor of two, while the strongest model velocities
5  change little. Observed convective available potential energy (CAPE) does not differ greatly between the two time periods, consistent with the small changes in median vertical velocities but not the larger changes in the strong tails of the distribution. These early results point to both opportunities and challenges in the development of new parameterization strategies. The excessively strong modeled median vertical velocities suggest examining alternate formulations for entrainment (de Rooy et al., 2013; Zhang et al., 2015; Lu et al., 2016), drawing on recent research in this area. The striking differences in how the median
10  and extreme velocities differ in the two time periods suggest more fundamental changes in the parameterization framework. The parameterization currently forms its plumes in the mean state. The similarity of CAPE during both time periods does not favor large differences in vertical velocities. The explanation could well be sub-grid variability in thermodynamic state, proba-

bly related to convective organization and currently not parameterized. Another important factor is a lack of scale awareness in the parameterization, which has been calibrated for scales comparable to those of the GATE campaign. Accounting for sub-grid variability and modeling the transition to explicit representation of these scales as resolution increases are related problems. In this perspective, we do not propose solutions to these issues but emphasize the importance of observations of vertical velocities

at convective scales to guide future modeling, explicit and parameterized, of convection in the climate system.

Wang et al. (2011) compared changes in shortwave cloud forcing from anthropogenic aerosols in CAM5 with these changes in a version of CAM5 in which a two-dimensional cloud model was used in place of CAM5's cloud and convection parameterizations. Wang et al. (2011)'s approach provides a distribution of sub-grid vertical velocities. They did not provide information on its characteristics or comparisons with observations, but it likely differs substantially from CAM5, which does not param-

eterize vertical velocities in its convection parameterization. It is not possible to assess how much of the 50% reduction in forcing using the cloud model in CAM5 is due to changes in sub-grid vertical velocities or even whether their effect has been buffered (reduced) by other processes. These changes in forcing related to sub-grid parameterization are larger than the changes associated with resolution changes discussed above, which together could comprise a large fraction of the forcing.

In the longer-term future, global models with resolutions fine enough to explicitly resolve deep cumulus clouds are envi-

sioned, eliminating the need for parameterizations of deep cumulus convection. Very short time integrations of models with horizontal resolutions of 0.87 km have already been reported (Miyamoto et al., 2013). In these models, and in limited-domain cloud-system models like those used in Wang et al. (2011), the preceding discussions imply a corresponding concern with the realism of their resolved vertical velocities. Evaluations of cloud-system models with horizontal resolutions as fine as 1 km show that many of these models produce vertical velocities that are too strong. These models show some similarities to

the parameterized velocities in Fig. 5 (Varble et al., 2014). Even in large-eddy simulations with resolutions on the order of tens of meters, important details of the distributions of vertical velocities are at variance with observations (Guo et al., 2008). Vertical velocities in these models can depend strongly on the method used to model their microphysics, and including more physically based microphysics in these models may improve the simulated vertical velocities (Fan et al., 2015). The methods used to model turbulence below even the fine resolutions in these models are also important and offer another path forward

(Bogenschutz and Krueger, 2013).

We have claimed that updraft speeds are among the controls on climate forcing and provide an observable constraint related to climate sensitivity. These relationships among vertical velocities, climate forcing, and climate sensitivity emerge through the effects of vertical velocities on physical processes. Resolution dependence of vertical velocities and limited attention to their sub-grid parameterization in models for climate and numerical weather prediction could consequently limit the realism of

these models. Taking account of these issues could open promising new paths toward reducing model uncertainties in climate forcing and sensitivity. Sub-grid parameterizations should include vertical velocities where the underlying physical processes depend on them. Some parameterizations already do so (e.g., Fig. 5; Donner (1993), Donner et al. (2011), and Chikira and Sugiyama (2010) for deep cumulus convection; Golaz et al. (2002) and Bretherton et al. (2004) for boundary-layer clouds).

Aerosol and cloud microphysical processes depend nonlinearly on vertical velocities, and physically based parameterizations which include these dependencies will fail if driven even by realistically averaged velocity fields, which may even smooth away such phenomena as small-scale updrafts necessary for aerosol activation. Until models explicitly resolve these scales, it is worth investigating simple scaling of resolved vertical velocities when using them to drive parameterizations. Fig. 3 implies that resolved vertical velocities will scale with a power-law dependence on resolution of $\Delta x^{-2/3}$, suggesting a scaling of vertical velocity for a physical parameterization $(\frac{\Delta x_{param}}{\Delta x})^{-2/3}$. $\Delta x_{param}$ is the scale at which a parameterization is physically realistic or becomes "scale aware" by design. For example, for a cloud system with uniform mesoscale ascent at a physical scale of around 10 km in a model with a resolution of 100 km, the vertical velocities for aerosol activation would be scaled from 100 km to 10 km. This approach would further have the advantage of requiring physical parameterizations to identify the scales at which their physics applies. It would also introduce at least a crude "scale awareness" into parameterizations which lack them. The vertical-velocity scaling also implies that resolved, advected fields, such as water vapor, will also have resolution dependencies. So, scaling vertical velocities used in parameterizations would introduce consistency issues not easily reconciled, underscoring the importance of pursuing resolutions as close as possible to the physical scales of climate processes.

## 5  Outlook and Challenges

The atmosphere sustains a broad spectrum of vertical motions. We posit that vertical velocities on all scales (deep and shallow convection, large eddies in stratiform clouds, large-scale ascent) carry important clues to climate forcing and climate sensitivity. The magnitudes of upward motions are an important control on the formation of liquid and ice particles in clouds and, consequently, anthropogenic climate forcing by cloud-aerosol interactions. In climate models, convective entrainment and mixing, which are among the key governors of vertical velocity, are related to climate sensitivity. Recent observations of convective vertical velocities could provide an important observational constraint for both anthropogenic climate forcing and climate sensitivity. Insightful analysis of these observations in the context of climate models could reduce two of the major uncertainties in climate change.

In climate models, both resolved and sub-grid vertical velocities are important. Modeling strategies should include (1) recognizing the dependence of vertical velocities on resolution and exploiting this dependence to scale resolved vertical velocities to process scales, (2) parameterizing sub-grid vertical velocities where the cloud-scale processes depend on them, and (3) explicitly taking into account scale and scale dependence for physical processes, both for resolved and parameterized processes. Preliminary observational studies and new approaches in parameterization are providing the means for doing so.

Neither high-resolution climate models, nor those with advanced parameterizations, will satisfactorily deal with connections among vertical velocities, climate forcing, and climate sensitivity if not grounded in realistic cloud-resolving and large-eddy models. Current cloud-resolving models require further development, based on preliminary comparison of their vertical velocities with observations. An intersection of resolution, microphysics, and turbulence will likely bring these models and observations into agreement. A high research priority is to focus on these issues in cloud-resolving and large-eddy modeling.

Observationally, field observations of atmospheric updrafts at cloud scale remain limited and should be expanded to sample a wider range of synoptic settings. If a high degree of confidence can be established in cloud-resolving and large-eddy models based on these observational studies, the models can be used to explore the many contexts in which cloud systems develop in the climate system.

Both anthropogenic climate forcing by aerosols and climate sensitivity are extremely difficult problems which have challenged climate scientists for decades. We raise the prospect here that new observations of an element of the climate system, its spectrum of updrafts on all scales, could provide important new clues. The argument that updrafts are a key to unlocking climate forcing and sensitivity is nuanced (especially for sensitivity). It is critical to determine the extent to which vertical velocities control climate forcing and constrain climate sensitivity. If vertical velocities were indeed to provide a breakthrough on this problem, they would do so through a satisfying unification of observational, theoretical, and modeling across the scales and phenomena that comprise the very broad field of contemporary atmospheric science.

*Acknowledgements.* Analysis of convective vertical velocities has been supported by the US Department of Energy Atmospheric System Research program through inter-agency agreement DE-SC0004534-NOAA. Reviews by Levi Silvers, Yi Ming, anonymous ACPD reviewers, as well as comments from Graham Feingold on controls on drop activation in warm clouds, are appreciated. Charles Seman participated in figure preparation.

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
