# Peer review of "Are Atmospheric Updrafts a Key to Unlocking Climate Forcing and Sensitivity?"

_Atmospheric Chemistry and Physics, 2016_

## Referee Comment (RC1) · Anonymous Referee #1 · 12 Jul 2016

This paper reviews basic issues around vertical velocity arising from the truncation of the spectrum in numerical models, and argues that this is affecting current estimates of aerosol forcing and possibly cloud feedback. The paper is a useful essay but I believe a few weakness should be addressed in a revised version.

1. The resolution-dependence of w has been examined by Pauluis and Garner 2006, which is not cited. I suggest shortening the text by invoking their results and analysis.

2. I found that the somewhat wordy paper was a bit light in terms of analysis. For example, can't the scaling laws quoted combined with the results in Fig. 1 be used to make a back-of-the-envelope estimate of how indirect effects should depend on resolution? A study is quoted showing a 30% decrease when changing

the resolution from 2 degrees to 1/4 degree, but there could be many reasons for this. It would make the argument more powerful if a simple calculation corroborated the magnitude of the impact based on the w-scaling argument. Also I think there are more studies that should be marshaled here, at least those of Wang et al. 2011 and more recent ones by Hugh Morrison.

Also, the discussion of how vertical velocity might be important to climate sensitivity was much less convincing than the discussion of aerosol effects. The role of the convective entrainment rate for example probably has more to do with altering the spectrum of convective depth, or how active the convective scheme is relative to large-scale condensation, than by altering w values per se. Sherwood et al. 2014 is invoked, but does that paper ever mention vertical velocity? Can the authors sharpen the reasoning here (for example, would we expect stronger updrafts and downdrafts to increase or decrease climate sensitivity?)

On the other hand, w seems important for overshooting of convective cells and mixing near the top of the cloud layer, which could be important for troposphere-stratosphere transport, gravity waves, etc. It also seems that the degree of localization of updrafts (hence precipitation) within a large grid cell for a given mass flux would be of interest for local weather and climate impacts.

3. A similar comment holds about solutions to the problem. The lead author Donner has already represented vertical velocities in a climate model parameterization, and to his credit reveals in Fig. 5 that the scheme does not capture any of the near-doubling of extreme updraft speed observed between the 19 Jan to 23 Jan field cases. Yet the paper says zero about this. Why did updrafts strengthen and why didn't the scheme pick this up? What physical assumptions went into the Donner 2011 scheme and do Donner et al. still think they are the right ones? Why or why not? The "Outlook and Challenges" section is largely just a summary of the points made earlier, then a laundry list of things people should be thinking about, chief among them that model physics be made scale-aware. Yet,

scale-awareness in a parameterizations is only helpful when one wants it to work automatically at different resolutions; presumably the Donner 2011 parameterization (for example) was optimized for the resolution of the GFDL model in which it was used, and was therefore already "aware" of the scale—yet still seems to lack a key sensitivity. What is it missing? What has been learned from the efforts so far? Leo is in as good a position as anyone to tell us and his insights would be useful here.

4. There was a lot of repetition in the paper, where a concept is explained in one place and then explained again when the manuscript comes back to it. I urge the authors to go through and try to organize it a bit better so that all the ideas are noted at the beginning but discussed in further detail in only one place in the text.

Fig. 1: In my pdf at least, this figure has multiple glitches that must be fixed—missing subscripts in figure notations, garbled title of lower panel, no labels on color legend, negative signs missing on y-axis labels.

p5,25: "relatedly" not an English word

p5,30: what is meant by "vertical transport"? The total net upward mass transport should not increase since it is governed by energy and mass balance, but there should be stronger localized upward (and downward) transports.

p6,5: Wang et al. 2011 also seems relevant here.

Fig. 4: why are numbers on the x-axis increasing to the left? The way this caption and figure are constructed means one has to think very hard to figure out the direction of the effect. If I have it right, one obtains more particles at finer resolution (as expected).

p8,3-10 this text mostly restates things said earlier.

p8,28-33 ditto

Fig. 5: In the caption please specify that each panel represents a different day and that

the colors represent three percentiles of w.

---

## Referee Comment (RC2) · Anonymous Referee #2 · 17 Jul 2016

General comment: This paper discusses two issues that are key to further advancing climate models, representation of updraft and scale-dependence, primarily in the context of reducing uncertainties in climate forcing and climate sensitivity. Highlighting the two understudied issues is timely, and thus I recommend its publication after the following concerns are addressed.

Specific comments:

1. The manuscript reads more like a "perspective" than a "review". It seems a bit thin as a review paper. As a perspective, it could be condensed by consolidating some repetitive texts. Maybe the authors have something in between in mind, based on the sentence "This review presents the perspective that ...." in P2. Guess the decision is between the editor and the authors.

[Figure]

2. It is not clear whether the paper is about deep convection, shallow convection, or convection in general. Please clarify. In fact, the issues in question are very generic.

3. The scaling argument in the second paragraph of Section 4 seems incomplete. The resolution dependence derived from the continuity equation should be for vertical velocity difference, not the velocity itself. A relationship between the vertical velocity and its spatial difference seems necessary for the latter?

4. The discussion on role of vertical velocity in climate sensitivity in Section 3 is not as obvious as one would like. Entrainment and convective mixing are identified; but some link of entrainment and mixing with vertical velocity would help. For example, a recent study by Lu et al (J. Atmos. Sci. 73, 761-773, DOI: 10.1175/JAS-D-15-0050) examined the relationship between vertical velocity and entrainment rate in shallow cu.

5. Guo et al (Characteristics of vertical velocity in marine stratocumulus: Comparison of LES simulations with observations, Environ. Res. Lett., 3, 0450J. doi:10.1088/1748-9326/3/4/045020) seems a good ref to the discussion in P8, esp. in the context of how well vertical velocity is represented in LES, its PDF, structure function and resolution dependence.

6. Some subscripts are missing in Fig 1, maybe due to file conversion. Also what do the different colors represent?

---

## Author Comment (AC1) · 18 Aug 2016

**Response to Referee #1**

We thank the referee for these comments and suggestions.

We have italicized the referee comments and bold-faced material which has been added in the revised manuscript. The revised manuscript is provided as a supplement to this response.

*1. The resolution-dependence of w has been examined by Paulius and Garner 2006,*

*which is not cited. I suggest shortening the text by invoking their results and analysis.*

The resolution dependence discussed in this paper refers to climate models with resolutions of 20 km and coarser. Pauluis and Garner (2006) identify a resolution dependence that applied for resolutions of 16 km and finer and show their scaling does not apply for coarser resolutions. Their scaling applied to a non-hydrostatic radiative-convective model and was not observationally confirmed. The scaling we discuss applies at scales that are largely hydrostatic and is consistent with Cho and Lindborg's (2001) observations. The scaling we discuss depends on the model grid spacing and is general over many types of motions, while Pauluis and Garner's (2006) applies to convective updrafts and depends on grid spacing normalized by updraft extent. The exponents in the power laws for the scalings reported here and those Pauluis and Garner (2006) present differ. Since the resolution dependencies differ in fundamental ways, we cannot use Pauluis and Garner (2006) to illustrate issues related to scaling of vertical velocities resolved by dynamical cores used for climate models. It's important to note that the scalings we report may not extend to the resolutions explored by Pauluis and Garner (2006), so we have added the following in Section 4:

**At considerably smaller, non-hydrostatic scales, the nature of the scaling may change. Pauluis and Garner (2006) report that updraft speeds in a non-hydrostatic model scale with the ratio of grid size to updraft vertical extent for resolutions finer than 16 km.**

*2. I found that the somewhat wordy paper was a bit light in terms of analysis. For example, can't the scaling laws quoted combined with the results in Fig. 1 be used to make a back-of-the-envelope estimate of how indirect effects should depend on resolution? A study is quoted showing a 30% decrease when changing the resolution from 2 degrees to 1/4 degree, but there could be many reasons for this. It would make*

*the argument more powerful if a simple calculation corroborated the magnitude of the impact based on the w-scaling argument. Also I think there are more studies that should be marshaled here, at least those of Wang et al. 2011 and more recent ones by Hugh Morrison.*

*Also, the discussion of how vertical velocity might be important to climate sensitivity was much less convincing than the discussion of aerosol effects. The role of the convective entrainment rate for example probably has more to do with altering the spectrum of convective depth, or how active the convective scheme is relative to large-scale condensation, than by altering w values per se. Sherwood et al. 2014 is invoked, but does that paper ever mention vertical velocity? Can the authors sharpen the reasoning here (for example, would we expect stronger updrafts and downdrafts to increase or decrease climate sensitivity?)*

*On the other hand, w seems important for overshooting of convective cells and mixing near the top of the cloud layer, which could be important for troposphere-stratosphere transport, gravity waves, etc. It also seems that the degree of localization of updrafts (hence precipitation) within a large grid cell for a given mass flux would be of interest for local weather and climate impacts.*

We have added the following text, which includes a "back-of-the-envelope" estimate of the effect of resolution dependence of vertical velocity on climate forcing by aerosol-cloud interactions, to Section 4. This estimate agrees well with the change in forcing obtained by Ma et al. (2015), discussed immediately thereafter in the text.

**Stevens (2015) relates climate forcing $F_{aci}$ by aerosol-cloud interactions to an-**

thropogenic changes in cloud drop number $N_d$ :

$$F_{aci} = -CE\left(\frac{\delta N_d}{N_d}\right), \tag{1}$$

where $CE$ is the effective cloud fraction. All quantities are global, annual means. A very rough estimate of the effect of the resolution dependence of vertical velocity can be made assuming $CE$ is fixed and that the partial derivatives shown in Fig. 1 do not co-vary significantly, so that $\delta N_d$ is also fixed. For a resolution refinement from $2^{\mathbf{o}}$ to $0.25^{\mathbf{o}}$, the vertical velocity scalings in Fig. 3 and the values for the variation of drop number with vertical velocity in Fig. 1 imply a reduction in $F_{aci}$ of about 18% to 36%, due to increases in $N_d$ of about 22% to 56%. Larger vertical velocities increase the pre-industrial droplet number $N_d$ and reduce the sensitivity of clouds to the number perturbation $\delta N_d$.

An attempt to infer the mechanisms by which resolution changes in Ma et al. (2015) change aerosol forcing is well beyond the scope of this perspective. We have added text noting multiple mechanisms are associated with the forcing changes in Ma et al.'s (2015) resolution experiments. We also add here a discussion of Zhang et al. (2016):

Aerosol forcing in a comprehensive model like that used by Ma et al. (2015) depends not only on droplet and ice nucleation and their relationships to vertical velocity. Ma et al. (2015) attributed their resolution sensitivities for aerosol indirect forcing to the resolution dependencies of their parameterizations for droplet nucleation and precipitation. Despite this, the resolution sensitivity of forcing found by Ma et al. (2015) agrees well with the estimate based on (1).

Zhang et al. (2016) examined an important component of aerosol indirect forcing, the sensitivity of liquid water path to the concentration of cloud conden-

**sation nuclei. They found this sensitivity to vary with dynamical regime within models and across models for individual regimes. Zhang et al. (2016)'s study is consistent with an important control on aerosol-cloud interactions by vertical velocities, which change with dynamical regime and, in all likelihood, within regimes across models.**

Wang et al. (2011) is very relevant but deals with sub-grid vertical velocities not resolved by the dynamical core of a climate model. So, we discuss it later in the section, after introducing the importance of sub-grid parameterizations:

**Wang et al. (2011) compared changes in shortwave cloud forcing from anthropogenic aerosols in CAM5 with these changes in a version of CAM5 in which a two-dimensional cloud model was used in place of CAM5's cloud and convection parameterizations. Wang et al. (2011)'s approach provides a distribution of sub-grid vertical velocities. They do not provide information on its characteristics or comparisons with observations, but it likely differs substantially from CAM5, which does not parameterize vertical velocities in its convection parameterization. It is not possible to assess how much of the 50% reduction in forcing using the cloud model in CAM5 is due to changes in sub-grid vertical velocities or even whether their effect has been buffered (reduced) by other processes. These changes in forcing related to sub-grid parameterization are larger than the changes associated with resolution changes discussed above, which together comprise a large fraction of the forcing.**

The reference to Sherwood et al. (2014) is intended to provide the reader with a possible explanation as to how convection could be a strong control on climate sensitivity. We offer this in light of the general consensus that low- and mid-level clouds are the most important source of uncertainty in climate sensitivity. Sherwood et al. (2014)

provide a possible mechanism linking convection with low- and mid-level clouds. Sherwood et al. (2014) is not intended to make the case for the importance of vertical velocities directly. We do feel it is important to establish the possible role of convection in climate sensitivity as a part of our argument that convective vertical velocities may provide critical clues for climate sensitivity. At this stage, we do not have a sense as to whether stronger updrafts would increase or decrease climate sensitivity. Zhao (2014) shows that increasing entrainment, with other factors held fixed in a GCM under development at GFDL, reduces sensitivity. For a plume in a specified environment, this would suggest stronger updrafts might be associated with increased sensitivity, but the generality of this result and the precise mechanisms involved are uncertain. Given the multiplicity of ways in which convection could influence climate sensitivity, it's very possible there is not a uniform response to stronger vertical velocities. Rather, response could depend on cloud regime and location, with the overall effect on sensitivity a composite of varying responses.

The vertical velocities (and entrainment rate controls on sensitivity in models) are indicators of processes important for sensitivity, not direct drivers of a process in the way that they are for activation. We agree with the referee and have made this point explicitly with the following addition to Section 3:

**The mechanisms discussed above explore fundamental characteristics of convection (convective mixing with associated de-hydration of low-cloud layers, shape and vertical extent of convective heating and moistening, convective microphysics, interactions between convective and stratiform precipitation) and their possible relationships to climate sensitivity. Vertical velocity does not directly relate to climate sensitivity, but, rather, correlates with these characteristics and is an indicator of how they are functioning in the climate system. As observed vertical velocities become available at convective scale, they thereby**

**provide an important, previously unrealized, constraint on these processes.**

*3. A similar comment holds about solutions to the problem. The lead author Donner has already represented vertical velocities in a climate model parameterization, and to his credit reveals in Fig. 5 that the scheme does not capture any of the near-doubling of extreme updraft speed observed between the 19 Jan to 23 Jan field cases. Yet the paper says zero about this. Why did updrafts strengthen and why didn't the scheme pick this up? What physical assumptions went into the Donner 2011 scheme and do Donner et al. still think they are the right ones? Why or why not? The "Outlook and Challenges" section is largely just a summary of the points made earlier, then a laundry list of things people should be thinking about, chief among them that model physics be made scale-aware. Yet, scale-awareness in a parameterizations is only helpful when one wants it to work automatically at different resolutions; presumably the Donner 2011 parameterization (for example) was optimized for the resolution of the GFDL model in which it was used, and was therefore already "aware" of the scale-yet still seems to lack a key sensitivity. What is it missing? What has been learned from the efforts so far? Leo is in as good a position as anyone to tell us and his insights would be useful here.*

We have discussed these issues by adding the following to the text in Section 4:

**Consistent with radar observations, the modeled median vertical velocities are similar over both time periods analyzed, but the observed strongest 1% of the vertical velocities differ by about a factor of two, while the strongest model velocities change little. Observed convective available potential energy (CAPE) does not differ greatly between the two time periods, consistent with the small changes in median vertical velocities but not the larger changes in the strong tails of the distribution. These early results point to both opportunities and chal-**

lenges in the development of new parameterization strategies. The excessively strong modeled median vertical velocities suggest examining alternate formulations for entrainment (de Rooy et al., 2013; Zhang et al., 2015; Lu et al., 2016), drawing on recent research in this area. The striking differences in how the median and extreme velocities differ in the two time periods suggest more fundamental changes in the parameterization framework. The parameterization currently forms its plumes in the mean state. The similarity of CAPE during both time periods does not favor large differences in vertical velocities. The explanation could well be sub-grid variability in thermodynamic state, probably related to convective organization and currently not parameterized. Another important factor is a lack of scale awareness in the parameterization, which has been calibrated for scales comparable to those of the GATE campaign. Accounting for sub-grid variability and modeling the transition to explicit representation of these scales as resolution increases are related problems. In this perspective, we do not propose solutions to these issues but emphasize the importance of observations of vertical velocities at convective scales to guide future modeling, explicit and parameterized, of convection in the climate system.

*4. There was a lot of repetition in the paper, where a concept is explained in one place and then explained again when the manuscript comes back to it. I urge the authors to go through and try to organize it a bit better so that all the ideas are noted at the beginning but discussed in further detail in only one place in the text.*

We have intentionally emphasized key points by repetition and would prefer as a matter of style to retain this. We do recognize style preferences vary on such matters and appreciate the reviewer's comment on this point but request forbearance regarding our differences in stylistic preference here.

*Fig. 1: In my PDF at least, this figure has multiple glitches that must be fixed: missing subscripts in figure notations, garbled title of lower panel, no labels on color legend, negative signs missing on y-axis labels.*

We do not have these problems on our PDF versions, but it does appear to depend on how you open the document from http://www.atmos-chem-phys-discuss.net/acp-2016-400/#discussion. If opened embedded inside the Firefox browser, the labels and subscripts are correct. If the very same document is downloaded and opened within Adobe Acrobat Reader DC it is as messed up as you described it.

We will work with ACP production staff to resolve any remaining problems with the PDF.

*p5,25: "relatedly" not an English word*

Dictionary.com and Collinsdictionary.com both define "relatedly" as an adverb, with Collins showing a general trend of its increased use over much of the last decade.

*p5,30: what is meant by "vertical transport"?  The total net upward mass transport should not increase since it is governed by energy and mass balance, but there should be stronger localized upward (and downward) transports.*

p5,30: We have changed the words "vertical transport" to "vertical mass flux", which hopefully conveys our meaning more clearly. Globally, energy and mass balances hold, as the reviewer notes, but localized changes in the upward and downward mass fluxes, which comprise the global balance, are important.

*p6,5: Wang et al. 2011 also seems relevant here.*

Wang et al. (2011) is now discussed; see the response to comment 2. It's worth noting an important difference, though, in the relevance of Ma et al. (2015) and Wang et al. (2011) in the context of this discussion about *resolved* vertical velocities, as opposed to parameterized vertical velocities. Both parameterized and resolved upward motions are important for aerosol activation. The point we are making in this section is that changing model resolution changes aerosol-cloud forcing and associated microphysics, which are precisely the experiments reported by Ma et al. (2015) and with ICON-ART in this paper. Wang et al. (2011) report work at a single model resolution, with the effect of vertical velocity entering in the (super-) parameterization they use, i.e., a two-dimensional cloud resolving model. The assertions in this paper on model resolution predict that two simulations at different model resolutions, both parameterizing cloud-aerosol interactions using an embedded two-dimensional cloud model, would exhibit different cloud-aerosol forcing, just as those using the standard CAM5 parameterizations do in Ma et al. (2015), though likely not quantitatively the same.

*Fig. 4: why are numbers on the x-axis increasing to the left? The way this caption and figure are constructed means one has to think very hard to figure out the direction of the effect. If I have it right, one obtains more particles at finer resolution (as expected)*

We have rearranged the figure with increasing numbers to the right. Additionally, we have added a small remark to the caption providing more guidance for the reader. Addition to the caption:

**Values of $R_{\Delta x_2, \Delta x_1} > 0$ ($< 0$) indicate an increase (a decrease) in ice crystal nucleation with an increase in resolution.**

*p8,3-10 this text mostly restates things said earlier.*

*p8,28-33 ditto*

p8,3-10, and p8, 28-33: See response to comment 4 above.

*Fig. 5: In the caption please specify that each panel represents a different day and that the colors represent three percentiles of w.*

Fig. 5 caption revised:

[revised manuscript text omitted]

---

## Author Comment (AC2) · 18 Aug 2016

**Response to Referee #2**

We thank the referee for these comments and suggestions.

We have italicized the referee comments and bold-faced material which has been added to the revised manuscript. The revised manuscript is provided as a supplement to this response.

*1. The manuscript reads more like a "perspective" than a "review". It seems a bit thin*

[Figure]

*as a review paper. As a perspective, it could be condensed by consolidating some repetitive texts. Maybe the authors have something in between in mind, based on the sentence "This review presents the perspective that ...." in P2. Guess the decision is between the editor and the authors.*

The referee is correct. Our goal is to present a perspective. As such, we do not attempt to present a classical review but to present more selectively key literature and a few original results, intended to stimulate new lines of research going forward. We have also adopted a style more appropriate for a perspective, including emphasizing key points by recapitulating them after they are first explored and then their implications are further developed.

One of us (Bernhard Vogel) contacted editor Ulrich Pöschl regarding the suitability of a perspective for ACP, which does not formally have this category of paper at the present time. We have followed his suggestion in submitting it as a review but with the request that it be viewed as a perspective.

*2. It is not clear whether the paper is about deep convection, shallow convection, or convection in general. Please clarify. In fact, the issues in question are very generic.*

We do indeed intend that the issues we raise be regarded as generic, applying to vertical velocities on all scales. We have added the following explicit statement in Section 5:

**We posit that vertical velocities on all scales (deep and shallow convection, large eddies in stratiform clouds, large-scale ascent) carry important clues to climate forcing and climate sensitivity.**

*3. The scaling argument in the second paragraph of Section 4 seems incomplete. The resolution dependence derived from the continuity equation should be for vertical velocity difference, not the velocity itself. A relationship between the vertical velocity and its spatial difference seems necessary for the latter?*

If the continuity equation is integrated upward from the surface, where (approximately) vertical motion vanishes, the scaling presented here holds. The argument presented is a scaling only. Precisely, as the reviewer suggests, horizontal variations in horizontal velocity are related to vertical differences in vertical velocity. The pressure difference can be considered as part of the proportionality in the velocity scaling on p. 5, l. 18.

*4. The discussion on role of vertical velocity in climate sensitivity in Section 3 is not as obvious as one would like. Entrainment and convective mixing are identified; but some link of entrainment and mixing with vertical velocity would help. For example, a recent study by Lu et al (J. Atmos. Sci. 73, 761-773, DOI: 10.1175/JAS-D-15-0050) examined the relationship between vertical velocity and entrainment rate in shallow cu.*

We now explicitly note the link between entrainment and vertical velocity at the beginning of Section 3, where studies relating entrainment in climate models to their sensitivities are discussed:

**The strongest suggestions of a link emerge from several studies showing that convective entrainment, an important control on vertical velocity, is related to the climate sensitivity in general circulation models...**

In Section 4, where the revised manuscript discusses in more detail the modeled and observed convective vertical velocities in Fig. 5, we discuss the possibility that new

formulations for convective entrainment, including Lu et al. (2016), will change distributions of parameterized vertical velocities:

**Consistent with radar observations, the modeled median vertical velocities are similar over both time periods analyzed, but the observed strongest 1% of the vertical velocities differ by about a factor of two, while the strongest model velocities change little. Observed convective available potential energy (CAPE) does not differ greatly between the two time periods, consistent with the small changes in median vertical velocities but not the larger changes in the strong tails of the distribution. These early results point to both opportunities and challenges in the development of new parameterization strategies. The excessively strong modeled median vertical velocities suggest examining alternate formulations for entrainment (de Rooy et al., 2013; Zhang et al., 2015; Lu et al., 2016), drawing on recent research in this area. The striking differences in how the median and extreme velocities differ in the two time periods suggest more fundamental changes in the parameterization framework. The parameterization currently forms its plumes in the mean state. The similarity of CAPE during both time periods does not favor large differences in vertical velocities. The explanation could well be sub-grid variability in thermodynamic state, probably related to convective organization and currently not parameterized. Another important factor is a lack of scale awareness in the parameterization, which has been calibrated for scales comparable to those of the GATE campaign. Accounting for sub-grid variability and modeling the transition to explicit representation of these scales as resolution increases are related problems. In this perspective, we do not propose solutions to these issues but emphasize the importance of observation of vertical velocities at convective scales to guide future modeling, explicit and parameterized, of convection in the climate system.**

*5. Guo et al (Characteristics of vertical velocity in marine stratocumulus: Comparison of LES simulations with observations, Environ. Res. Lett., 3, 0450J. doi:10.1088/1748-9326/3/4/045020) seems a good ref to the discussion in P8, esp. in the context of how well vertical velocity is represented in LES, its PDF, structure function and resolution dependence.*

This is an important reference, and we thank the reviewer for reminding us of it. We added the following in Section 4:

**Even in large-eddy simulations with resolutions on the order of tens of meters, important details of the distributions of vertical velocities are at variance with observations (Guo et al., 2008).**

*6. Some subscripts are missing in Fig 1, maybe due to file conversion. Also what do the different colors represent?*

The subscripts and a legend box showing the meaning of the colors appear on our PDF versions of the manuscript. Their doing so may depend on how you open the document from http://www.atmos-chem-phys-discuss.net/acp-2016-400/#discussion. If opened embedded inside the Firefox browser, the labels and subscripts are correct. If the very same document is downloaded and opened within Adobe Acrobat Reader DC there are problems of the nature the reviewer describes.

We will work with ACP production staff to resolve any remaining problems with the PDF.